# Availability of adequately iodized salt at the household level in Ethiopia: A systematic review and meta-analysis

**Alehegn Aderaw Alamneh**[1]*, **Cheru Tesema Leshargie**[2,3], **Melaku Desta**[4], **Molla Yigzaw Birhanu**[5], **Moges Agazhe Assemie**[2], **Habtamu Temesgen Denekew**[1], **Yoseph Merkeb Alamneh**[6], **Daniel Bekele Ketema**[2]

1 Department of Human Nutrition and Food Sciences, College of Health Sciences, Debre Markos University, Debre Markos, Ethiopia, 2 Department of Public Health, College of Health Sciences, Debre Markos University, Debre Markos, Ethiopia, 3 Faculty of Health, University of Technology Sydney, Sydney, Australia, 4 Department of Midwifery, College of Health Sciences, Debre Markos University, Debre Markos, Ethiopia, 5 College of Health Sciences, Debre Markos University, Debre Markos, Ethiopia, 6 Departments of Biomedical Sciences, Schools of Medicine, Debre Markos University, Debre Markos, Ethiopia

* alehegn12aderaw@gmail.com

**Data Availability Statement:** All relevant data are within the manuscript and its Supporting Information files.

## Abstract

### Background

Iodine deficiency disorder (IDD) is a global, regional, and national public health problem that is preventable. Universal salt iodization is a worldwide accepted strategy to prevent IDD. The level of iodine in the salt should be adequate at the household level ($\geq$15ppm). Though there was fragmented evidence on the proportion of adequately iodized salt at the household level in Ethiopia, the national level proportion of adequately iodized salt at the household level was remaining unknown. Therefore, this systematic review and meta-analysis estimated the pooled proportion of adequately iodized salt at the household level in Ethiopia from 2013–2020.

### Method

We systematically searched the databases: PubMed/MEDLINE, Google Scholar, and Science Direct for studies conducted in Ethiopia on the availability of adequately iodized salt at the household level since 2013. We have included observational studies, which were published between January first, 2013, and 10 August 2020. The report was compiled according to the Preferred Reporting Items for Systematic Reviews and Meta-Analysis (PRISMA) guidelines. The quality of included studies was scored based on the Newcastle Ottawa quality assessment scale adapted for cross-sectional studies. The data were extracted in Microsoft excel and analyzed using Stata version 14.1 software. We employed a random-effects model to estimate the pooled proportion of adequately iodized salt at the household level in Ethiopia. The presence of statistical heterogeneity within the included studies was evaluated using the I-squared statistic. We used Egger's regression test to identify evidence of publication bias. The pooled proportion with a 95% confidence interval (CI) was presented using tables and forest plots.

**Funding:** The authors received no specific funding for this work.

**Competing interests:** The authors have declared that no competing interests exist.

**Abbreviations:** AIS, Adequately Iodized Salt; CI, Confidence Interval; HH, Household; ICCIDD, International Council for Control of Iodine Deficiency Disorders; IDD, Iodine Deficiency Disorders; NOQS, Newcastle Ottawa Quality Score; PPM, Parts Per Million; SNNPR, Southern Nations Nationalities and Peoples Region; USI, Universal Salt Iodization; WHO, World Health Organization.

## Results

We screened a total of 195 articles. Of these, 28 studies (with 15561 households) were included in the final systematic review and meta-analysis. In Ethiopia, the pooled proportion of adequately iodized salt at the household level was 37% (95% CI: 28, 46%). The subgroup analyses of 28 studies by residence revealed that the pooled proportion of adequately iodized salt at the household level was 32% (95% CI: 29, 35%) and 48% (95% CI: 31, 66%) in rural and urban areas, respectively. Based on geographic location, the highest proportion was found in Addis Ababa (81%; 95%CI: 78, 83), and the lowest proportion found in Dire Dawa (20%; 95%CI: 17, 22). Besides, the proportion of adequately iodized salt at the household level was significantly increased during 2017–2020 (42%; 95% CI: 30, 53%) as compared with 2013–2016 (27%; 95% CI: 17, 39%).

## Conclusions

In Ethiopia, the pooled proportion of adequately iodized salt at the household level was very low as compared to the world health organization's recommendation. Thus, the Federal Ministry of Health of Ethiopia and different stakeholders should give more attention to improve the proportion of adequately iodized salt at the household level.

## Background

Iodine is a chemical element that is essential for the synthesis of thyroid hormone by the thyroid gland in the body. Thyroid hormones are essential for the normal development and function of the brain and nervous system, and the maintenance of body heat and energy. When people do not have enough iodine, they cannot make enough thyroid hormone. This deficiency of iodine has several important health consequences that together are called iodine deficiency disorders (IDD). Iodine deficiency frequently causes permanent brain damage and cognitive impairment in children, reproductive failure (miscarriages, stillbirths), decreased child survival, goiter, and socioeconomic stagnation. Iodine deficiency is important because of its widespread prevalence and its destructive effects on human health. Proper supplementation with iodine completely prevents these consequences. Iodine is supplemented in the form of iodized salt, iodized oil, iodized water, and frequent administration of Lugol's iodine. Among these, salt iodization has been proven and the most effective strategy to prevent IDD at the population level [1, 2].

Iodine deficiency is also a public health important problem in Ethiopia. The national total goiter rate among Ethiopian women was above 35.8% [3]. Also, the pooled estimate of goiter among children in Ethiopia was 40.50% Thus, the government of Ethiopia recommended and implemented universal salt iodization (USI) to prevent iodine deficiency and its associated deficiency disorders [4, 5]. The availability of adequately iodized salt at the household level is one of the process indicators used to monitor the consumption of iodized salt at the population level. According to the World Health Organization recommendation, the coverage of adequately iodized salt at the household level should be above 90% to prevent iodine deficiency disorders [2].

In Ethiopia, the proportion of adequately iodized salt at the household level has been reported in several studies, which is inconsistent and ranges from 4.6% at Dega Damot Districts of Amhara region [6] to 95.5% at Kolfe Keranio sub-city of Addis Ababa [7]. As a result

of variations of findings across previously existing studies, producing a pooled proportion of adequately iodized salt at the household level is needed. Therefore, this systematic review and meta-analysis were conducted to produce the pooled proportion of adequately iodized salt at the household level in Ethiopia since 2013. The pooled estimate of adequately iodized salt at the household level will be an important indicator for the government, programmers, policy-makers, and different stakeholders to monitor the progress of adequately iodized salt coverage at the household level.

## Methods

### Data source and search strategy

The studies were found through internet searches using databases of PubMed, Google Scholar, and Science direct. Searching of the articles was done by AAA, DBK, MD, CTL, MAA, MYB & HTD using the keywords of "Availability", "Adequately Iodized salt" "Household Level" "Ethiopia" in combination or individually. The last search was conducted on 10 August 2020.

### Inclusion criteria

**Study setting.** Studies conducted in Ethiopia were included.

**Study units.** Studies conducted on the availability of adequately iodized salt at the household's level.

**Publication status.** Both published and unpublished articles were included.

**Language.** Only studies published in the English language were included.

**Study type.** Studies employed using observational study designs were included.

**Publication year.** Articles that were published between first January 2013 and 10th August 2020 were included. The rationale for including those studies published since January 2013 was to generate more recent information that will be useful for decision making.

**Type of article.** Only full-text articles were included.

### Exclusion criteria

Studies that did not report the outcome of interest and studies with the unsatisfactory quality score (Newcastle Ottawa quality score ≤4) were excluded from this systematic review and meta-analysis [8].

### Screening, data extraction, and quality assessment

Before conducting data abstraction, the data extraction format was prepared in a Microsoft™ Excel spreadsheet. The data extraction sheet includes the author's name, year of publication, study design, region, study area, residence sample size, response rate, and proportion of adequately iodized salt at the household level. Studies that fulfill the inclusion criteria were screened and extracted by AAA, DBK, MYB, CTL, MD, MAA, & YMA using the pre-defined data extraction format. Then, the two authors (AAA, DBK) done quality assessment independently for the included studies using the Newcastle-Ottawa Quality assessment scale adapted for cross-sectional studies. The quality assessment scale includes representativeness of the sample, sample size satisfactoriness, non-response rate, and validity of measurement tool, comparability of subjects in different outcome groups, outcome assessment, and statistical test [8]. The 2 reviewers each (AAA and DBK) scored the included articles based on the above-mentioned quality assessment criteria. The combined quality assessment score for each study ranges from 0–10. The two researchers who extracted the data were discussed to solve any

disagreements on data extractions under the mediator of the third author (YMA). Besides, the Microsoft Word PRISMA 2009 checklist was used to compile the report [9] (S1 File).

## Outcome measurement

Adequately iodized salt at household level: If a household salt is fortified with the iodine content of ≥15 parts per million (ppm).

## Statistical analysis

The data were extracted in excel and exported into Stata version 14 for analysis. The pooled estimate was computed using the "metaprop" command [10]. The original articles were described using forest plots and tables. There was statistically significant heterogeneity among studies. Therefore, we used a random-effect model to pool the proportion of adequately iodized salt at the household level. The pooled proportion with a 95% confidence interval was reported. Sub-group analysis was done by geographic location where the study was done, residence, year of publication, and sample size. Sensitivity analysis was done to check the influence of small studies on the pooled prevalence [11].

## Heterogeneity test and publication bias

The presence of statistical heterogeneity within the included studies was evaluated using the I-squared statistic. The heterogeneity was classified as low, medium, and high when the value of I-squared was around 25%, 50%, and 75%, respectively [12]. We used Egger's regression test to identify evidence of publication bias. Statistically significant publication bias was declared at a p-value of less than 0.05. The trim and fill analysis was done to quantify the effect sizes of missed studies [13].

# Results

## Search results

A total of 195 studies were identified by the electronic search in PubMed, Google Scholar, and Science direct. Of which, 5 articles were excluded due to duplication, 161 were excluded based on the exclusion criteria, 1 study was excluded since they did not report the outcome of interest [14]. Finally, 28 cross-sectional studies were found to be eligible and included in the current systematic review and meta-analysis (Fig 1).

## Characteristics of reviewed studies

As shown in Table 1, a total of 28 studies (with 15561 households) met the inclusion criteria. Six regions and 2 city administrations were represented by this systematic review and meta-analysis. These are, 9 were Amhara region (n = 9) [6, 15–22], Oromia region(n = 8) [23–30], SNNPR (n = 4) [31–34], Tigray region (n = 2) [35, 36], Dire Dawa (n = 2) [37, 38], Addis Ababa (n = 2) [7, 39], and Benishangul Gumuz (n = 1) [40]. The smallest sample size (269) was reported from a study at the Sidama zone in SNNPR and the highest (1194) was from a study at the Dera district in the Amhara region. The quality score ranges from 7–10 with a quality score of good and very good. The proportion of adequately iodized salt at the household level as reported from the primary studies ranged from 4.6% at Dega Damot Districts of Amhara region [6] to 95.5% at Kolfie Keranio sub-city of Addis Ababa (7) (Table 1).

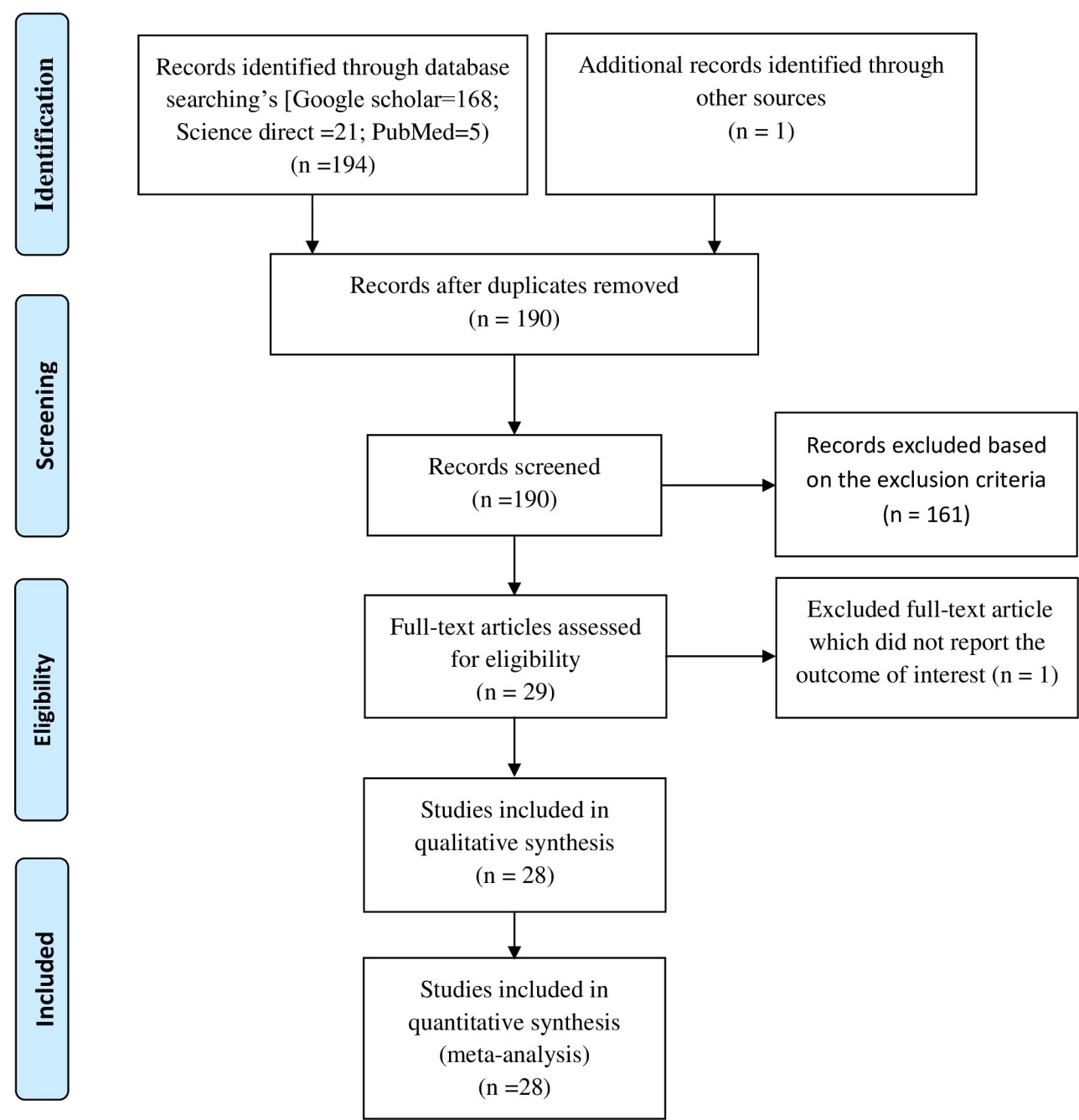

**Fig 1. PRISMA flow chart which shows the selection of included studies.**

## Pooled proportion of adequately iodized salt

The pooled proportion of adequately iodized salt at the household level was 37% (95% CI: 28, 46%; $I^2$ = 99.28%, p<0.001) in Ethiopia (Fig 2).

The subgroup analyses of 28 studies by residence revealed that the pooled proportion of adequately iodized salt at a household level among rural and urban residents was 32% (95% CI: 29, 35%) and 48% (95% CI: 31, 66%), respectively. Based on 28 included studies, the subgroup analysis of adequately iodized salt proportion at the household level by regions showed

**Table 1. Summary of the included studies which were done on the proportion of adequately iodized salt at HH level in Ethiopia, 2013–2020 (n = 28).**

| S. No. | Authors | Year of Publication | Region | Study Area | Study Setting | Sample size | Response Rate (%) | NOQS | Proportion HHs using AIS (%) |
|---|---|---|---|---|---|---|---|---|---|
| 1 | Mesele et al. [16] | 2014 | Amhara | Lay Armachiho | Both | 694 | 99.4 | 10 | 29.7 |
| 2 | Mekonnen et al. [19] | 2018 | Amhara | Dessie & Combolcha | Urban | 500 | 95.4 | 10 | 68.8 |
| 3 | Ajema et al. [31] | 2020 | SNNPR | Arba Minch | Urban | 875 | 100.0 | 10 | 58.20 |
| 4 | Abebe et al. [17] | 2017 | Amhara | Dabat | Both | 705 | 98.7 | 10 | 33.20 |
| 5 | Anteneh et al. [18] | 2017 | Amhara | Dera | Both | 1194 | 96.2 | 10 | 57.2 |
| 6 | Desta et al. [35] | 2019 | Tigray | Ahferom | Both | 292 | 91.8 | 8 | 17.5 |
| 7 | Gebriel et al. [40] | 2014 | Benishangul Gumuz | Assosa | Urban | 395 | 100.0 | 10 | 26.1 |
| 8 | Wondimagegn et al. [33] | 2018 | SNNPR | Wolaita Sodo | Both | 440 | 99.8 | 10 | 36.7 |
| 9 | Tariku et al. [20] | 2019 | Amhara | Mecha | Both | 700 | 98.0 | 10 | 63.3 |
| 10 | Gebremariam et al. [15] | 2013 | Amhara | Gondar | Urban | 810 | 95.5 | 10 | 28.9 |
| 11 | Hailu et al. [25] | 2016 | Oromia | Robe | Both | 393 | 93.1 | 10 | 29.0 |
| 12 | Gidey et al. [36] | 2015 | Tigray | Laelay Maychew | Rural | 600 | 98.4 | 9 | 33.0 |
| 13 | Yaye et al. [38] | 2016 | Dire Dawa | Dire Dawa | Urban | 694 | 100 | 10 | 7.5 |
| 14 | Hawas et al. [23] | 2016 | Oromia | Assela | Urban | 513 | 96.4 | 10 | 62.9 |
| 15 | Ayigegn et al. [7] | 2020 | Addis Ababa | Kolfie Keranio | Urban | 541 | 95.5 | 10 | 95.5 |
| 16 | Yazew [28] | 2020 | Oromia | Horro | Both | 390 | 100 | 8 | 23.6 |
| 17 | Meselech et al. [24] | 2016 | Oromia | Lalo Assabi | Both | 768 | 95.0 | 10 | 8.7 |
| 18 | Hiso et al. [29] | 2019 | Oromia | Duguda | Rural | 402 | 100 | 10 | 30.7 |
| 19 | Woyraw et al. [22] | 2018 | Amhara | Jabitehinan | Both | 549 | 98.0 | 9 | 48.3 |
| 20 | Aredo et al. [27] | 2020 | Oromia | Hetosa | Both | 596 | 98.8 | 8 | 61.1 |
| 21 | Tigabu et al. [21] | 2017 | Amhara | Gasgibla | Both | 443 | 97.6 | 10 | 17.2 |
| 22 | Asfaw et al. [32] | 2020 | SNNPR | Dewaro Zone | Both | 230 | NR | 7 | 19.1 |
| 23 | Fereja et al. [26] | 2018 | Oromia | Ada | Both | 351 | 98.3 | 10 | 39.3 |
| 24 | Afework et al. [6] | 2019 | Amhara | Dega Damot | Both | 802 | 100.0 | 10 | 4.6 |
| 25 | Ftwi et al. [37] | 2018 | Dire Dawa | Dire Dawa | Urban | 402 | 99.5 | 10 | 49.0 |
| 26 | Belay | 2020 | Addis Ababa | Kolfie Keranio | Urban | 417 | 98.5 | 10 | 63.8 |
| 27 | Stoecker et al. [34] | 2020 | SNNPR | Sidama Zone | Both | 269 | NR | 7 | 21.0 |
| 28 | Tololu et al. | 2016 | Oromia | Goba Town | Urban | 596 | 99.7 | 9 | 30.0 |

NOQS: Newcastle Ottawa Quality Score; HH: Household; AIS: Adequately Iodized Salt.

that the highest proportion was found in Addis Ababa (81%; 95%CI: 78, 83%) and the lowest was found in Dire Dawa (20%; 95%CI: 17, 22%). Also, the subgroup analysis of adequately iodized salt availability at the household level was done by the year of publication. The finding revealed that the proportion of adequately iodized salt at the household level was significantly increased during 2017–2020 (42%; 95% CI: 30, 53%) as compared with 2013–2016 (27%; 95% CI: 17, 39%) (Table 2).

## Meta-regression

We run a random effect meta-regression by year of publication, region, residence sample size, and quality score to detect the source of heterogeneity. The finding evidenced that there is a statistically significant variation of the proportion of adequately iodized salt at HH by year of publication and residence across the pooled studies (p <0.05). The proportion of between-

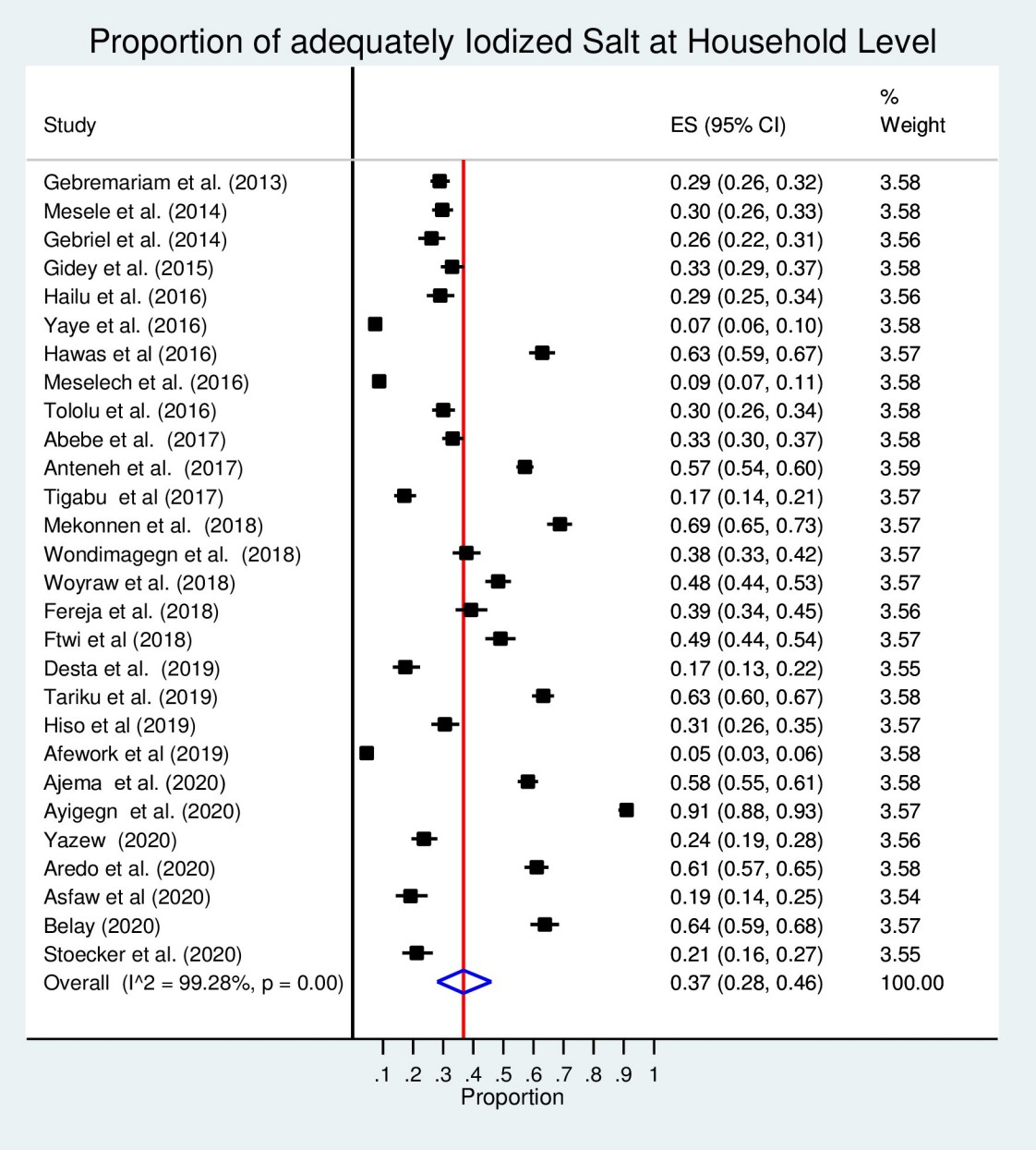

**Fig 2. Forest plot of the 28 included studies which assessed the proportion of adequately iodized salt at HHs level in Ethiopia, 2013–2020.**

study variation explained by year of publication, region, residence, sample size, and quality score was 29.23% (Table 3).

## Publication bias and sensitivity analysis

Funnel plot and Egger regression test methods were used to check publication bias. The finding evidenced asymmetrical funnel plot and statistically significant publication bias (p<0.05). The trim and fill analysis was done to quantify the effect sizes of missed studies. The finding showed that 10 studies with negative findings were missed from publishing. Besides, the

**Table 2. Subgroup analysis of the pooled proportion of adequately iodized salt at the HH level in Ethiopia by region, residence, & year of publication, 2013–2020.**

| Variables | Subgroup | No of included Studies | Sample size | Estimated proportion of AIS at HH level % (95% CI) |
|---|---|---|---|---|
| **Region** | Amhara | 9 | 6, 397 | 37 (23, 53) |
| | Oromia | 8 | 4, 009 | 35 (21, 50) |
| | SNNPR | 4 | 1, 814 | 33 (16, 53) |
| | Tigray | 2 | 892 | 28 (25, 31) |
| | Addis Ababa | 2 | 958 | 81 (78, 83) |
| | Dire Dawa | 2 | 1, 096 | 20 (17, 22) |
| | Benishangul Gumuz | 1 | 395 | 26 (22, 31) |
| **Residence** | Urban | 10 | 5, 743 | 48 (31, 66) |
| | Both | 16 | 8, 816 | 30 (21, 41) |
| | Rural | 2 | 1, 002 | 32 (29,35) |
| **Year of Publication** | 2013–2016 | 9 | 4, 867 | 27 (17, 39) |
| | 2017–2020 | 19 | 10, 694 | 42 (30, 53) |
| **Total** | | **38** | **15, 561** | **37 (28, 46)** |

sensitivity analysis finding showed that the individual studies did not have a significant impact on the overall pooled prevalence of adequately iodized salt at the household level.

## Discussion

This systematic review and meta-analysis finding showed that the pooled estimate of adequately iodized salt at the household level in Ethiopia was 37% (95% CI: 28, 46%; $I^2$ = 99.28%, p<0.001). The pooled estimate was varying by region, year of publication, and residence.

In Ethiopia, the pooled estimate of adequately iodized salt at the household level is low as compared to the world health organization's (WHO) recommendation. According to WHO recommendation, the proportion of households with adequately iodized salt should be more than 90% to prevent iodine deficiency disorders among the population [2]. This implies that the population in Ethiopia was exposed to iodine deficiency disorders.

Based on the geographic location where the studies conducted, the highest pooled estimate was found in Addis Ababa (81%; 95%CI: 78, 83%), and the lowest prevalence found in Dire Dawa (20%; 95%CI: 17, 22). This variation might be due to weather variation across the regions, which affects the level of iodine content [34].

The subgroup analysis of adequately iodized salt by year of publication showed that the proportion of adequately iodized salt at the household level was significantly increased during 2017–2020 (42%; 95% CI: 30, 53%) as compared with 2013–2016 (27%; 95% CI: 17, 39%). This finding is in line with the finding of a study conducted based on 10 national coverage surveys

**Table 3. Meta-regression of the proportion of AIS by year of publication, region, residence, sample size, & quality score to detect the source of heterogeneity in Ethiopia, 2013–2020 (n = 28).**

| Variable | Coefficient | p-value | 95% Conf. Interval |
|---|---|---|---|
| **Year of publication** | 5.298441 | 0.012* | 1.303038, 9.293845 |
| **Region** | -2.753277 | 0.235 | -7.433168, 1.926615 |
| **Residence** | 8.815902 | 0.039* | . .4790526, 17.15275 |
| **Sample size** | .0108135 | 0.581 | -.0291769, .0508039 |
| **Quality Score** | 3.172159 | 0.546 | -7.546717, 13.89104 |

*Statistically significant variation.

in 2016 [41]. These substantial increments might be due to the government and different stakeholder's efforts in enforcing USI laws and awareness creation on proper handling of iodized salt at the wholesaler, distributor, and household level.

The subgroup analysis of studies by residence revealed that the pooled prevalence of adequately iodized salt at the household level was higher among urban residents 48% (95% CI: 31, 66) as compared with the rural residents (32% (95% CI: 29, 35). This finding is also in line with the findings of a study conducted based on 10 national coverage surveys in 2016 [41]. Increased access to media and a high educational level in the urban area might be the possible explanations for this observed variation.

## Limitation of the study

The findings of this meta-analysis should be interpreted considering the following limitations. The first limitation is that this meta-analysis did not find a study from the two regional states of Ethiopia (Gambella and Afar) which limits the generalizability of the finding at the national level. Second, heterogeneity among the included studies was high ($I^2$ statistic = 99.28%, $p < 0.001$). Third, there is a statistically significant publication bias ($p > 0.05$). Hence, the random effect model was used to adjust the heterogeneity among the included studies. Also, meta-regression was done to identify the source of heterogeneity. The finding evidenced that year of publication and residence were the statistically significant variables introducing such a high variation among the included studies. Furthermore, trim and fill analysis was done to treat publication bias. The analysis indicated as 10 studies were missed.

## Conclusions

In conclusion, in Ethiopia, the pooled proportion of adequately iodized salt at the household level was very low as compared to the world health organization recommendation. This indicates that the population in Ethiopia was exposed to iodine deficiency disorders. Thus, the Federal Ministry of Health of Ethiopia and different stakeholders should give more attention to improve the proportion of adequately iodized salt at the household level.

## Supporting information

**S1 File. PRISMA flow 2009 checklist of the study.**
(DOCX)

**S2 File. Dataset.**
(DTA)

## Acknowledgments

We would like to acknowledge the authors of original articles in which without their work this systematic review and meta-analysis could not be conducted. At last but not least, we would like to acknowledge Mr. Fasil Wagnew, a Lecturer at Debre Markos University for his thorough copy edit of the manuscript for language usage, spelling, and grammar.

## Author Contributions

**Conceptualization:** Alehegn Aderaw Alamneh.

**Data curation:** Alehegn Aderaw Alamneh, Cheru Tesema Leshargie, Melaku Desta, Molla Yigzaw Birhanu, Moges Agazhe Assemie, Habtamu Temesgen Denekew, Yoseph Merkeb Alamneh, Daniel Bekele Ketema.

**Formal analysis:** Alehegn Aderaw Alamneh, Cheru Tesema Leshargie, Habtamu Temesgen Denekew, Daniel Bekele Ketema.

**Investigation:** Alehegn Aderaw Alamneh, Cheru Tesema Leshargie, Daniel Bekele Ketema.

**Methodology:** Alehegn Aderaw Alamneh, Cheru Tesema Leshargie, Melaku Desta, Molla Yigzaw Birhanu, Moges Agazhe Assemie, Habtamu Temesgen Denekew, Yoseph Merkeb Alamneh, Daniel Bekele Ketema.

**Project administration:** Alehegn Aderaw Alamneh.

**Resources:** Alehegn Aderaw Alamneh.

**Software:** Alehegn Aderaw Alamneh, Cheru Tesema Leshargie, Daniel Bekele Ketema.

**Supervision:** Alehegn Aderaw Alamneh, Habtamu Temesgen Denekew, Daniel Bekele Ketema.

**Validation:** Alehegn Aderaw Alamneh, Cheru Tesema Leshargie, Molla Yigzaw Birhanu, Habtamu Temesgen Denekew, Yoseph Merkeb Alamneh, Daniel Bekele Ketema.

**Visualization:** Alehegn Aderaw Alamneh, Melaku Desta, Moges Agazhe Assemie, Daniel Bekele Ketema.

**Writing – original draft:** Alehegn Aderaw Alamneh, Moges Agazhe Assemie.

**Writing – review & editing:** Alehegn Aderaw Alamneh, Cheru Tesema Leshargie, Melaku Desta, Molla Yigzaw Birhanu, Moges Agazhe Assemie, Habtamu Temesgen Denekew, Yoseph Merkeb Alamneh, Daniel Bekele Ketema.

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
