## [Decision Letter · Decision Letter 0]

7 Dec 2020

PONE-D-20-26914

Availability of adequately iodized salt at household level in Ethiopia: a systematic review and meta-analysis

PLOS ONE

Dear Dr. Alamneh,

Thank you for submitting your manuscript to PLOS ONE. After careful consideration, we feel that it has merit but does not fully meet PLOS ONE’s publication criteria as it currently stands. Therefore, we invite you to submit a revised version of the manuscript that addresses the points raised during the review process.

We look forward to receiving your revised manuscript.

Kind regards,

Shaun Wen Huey Lee, Ph.D.

Academic Editor

PLOS ONE

Journal Requirements:

2. Please provide your rationale for restricting search timeline to 2013 onward? Why have earlier publications been excluded?

3. Please use standard statistical reporting of p values. For example, please delete p=0.000 and replace with p <0.001.

4.We suggest you thoroughly copyedit your manuscript for language usage, spelling, and grammar. If you do not know anyone who can help you do this, you may wish to consider employing a professional scientific editing service.  

Additional Editor Comments (if provided):

The authors have written a well executed study and was in line with best practice. However, as highlighted by one of the reviewer, the authors ought to seek and use more inclusive languages in their manuscript as per PLOS guidelines.

Reviewers' comments:

Reviewer's Responses to Questions

**Comments to the Author**

1. Is the manuscript technically sound, and do the data support the conclusions?

Reviewer #1: Yes

Reviewer #2: Yes

2. Has the statistical analysis been performed appropriately and rigorously? 

Reviewer #1: Yes

Reviewer #2: Yes

3. Have the authors made all data underlying the findings in their manuscript fully available?

Reviewer #1: Yes

Reviewer #2: Yes

4. Is the manuscript presented in an intelligible fashion and written in standard English?

Reviewer #1: Yes

Reviewer #2: Yes

5. Review Comments to the Author

Reviewer #1: This paper is, for the most part, well written, straightforward, and easy to follow. Methods are appropriate and described adequately, and the statistical tests performed seem appropriate. However, it would be beneficial to read carefully through the manuscript to make sure everything is consistent and without typos. The topic is of importance.

Reviewer #2: I would recommend to edit and reformulate the background. The use if terms such as “mental retardation” and “fertility failure” are outdated and might even be considered offensive by vulnerable populations. I would recommend to substitue them with “cognitive impairment” and simply “infertility”, respectively.

Besides these remarks, I believe the data analysis can be of great relevance for the large scale nutritional interventions.

6. PLOS authors have the option to publish the peer review history of their article (what does this mean?). If published, this will include your full peer review and any attached files.

Reviewer #1: No

Reviewer #2: **Yes: **Heber Gomez-Malave

---

## [Author Response · Author response to Decision Letter 0]

16 Jan 2021

Response to Reviewers

Response to the Academic Editor

Thank you for your invaluable comments and suggestions. We have addressed the points you raised as follows: 

Response: Thank you! We have prepared the manuscript as per PLOS ONE guideline. 

2. Please provide your rationale for restricting search timeline to 2013 onward? Why have earlier publications been excluded?

Response: Dear academic editor, thank you for your concern. The rationale for restricting a search timeline to 2013 onward is to generate more recent information that will be useful for decision making (Check on Method section, page 5, lines 104 & 105). 

3. Please use standard statistical reporting of p values. For example, please delete p=0.000 and replace it with p <0.001.

Response: Thank you! We replaced it with p <0.001 (Check on page 10, lines 183; on page 13, line 221, and on page 14, line 250).

4. We suggest you thoroughly copyedit your manuscript for language usage, spelling, and grammar. If you do not know anyone who can help you do this, you may wish to consider employing a professional scientific editing service. 

Response: The manuscript was thoroughly copy edited for language usage, spelling, and grammar by our colleague Mr. Fasil Wagnew who is a Lecturer at Debre Markos University (Check on “Acknowledgements Section” page 17, lines 296-298; and see the “Revised Manuscript with Track Changes”).

Response to Reviewer #1

Reviewer #1: This paper is, for the most part, well written, straightforward, and easy to follow. Methods are appropriate and described adequately, and the statistical tests performed seem appropriate. However, it would be beneficial to read carefully through the manuscript to make sure everything is consistent and without typos. The topic is of importance.

Response: Dear Reviewer, thank you for your suggestion to read the manuscript carefully in order to make sure everything is consistent and without typos. Thus, we have read and revise the manuscript (Check on the “Revised Manuscript with Track Changes”. 

Response to Reviewer #2

Reviewer #2: I would recommend you to edit and reformulate the background. The use of terms such as “mental retardation” and “fertility failure” are outdated and might even be considered offensive by vulnerable populations. I would recommend you to substitute them with “cognitive impairment” and simply “infertility”, respectively. Besides these remarks, I believe the data analysis can be of great relevance for the large-scale nutritional interventions.

Response: Dear reviewer, thank you for your suggestion! We have edited those terms as per your suggestion (Check on page 3, line 64). 

.

---

## [Decision Letter · Decision Letter 1]

2 Feb 2021

Availability of adequately iodized salt at the household level in Ethiopia: a systematic review and meta-analysis

PONE-D-20-26914R1

Dear Dr. Alamneh,

We’re pleased to inform you that your manuscript has been judged scientifically suitable for publication and will be formally accepted for publication once it meets all outstanding technical requirements.

Kind regards,

Shaun Wen Huey Lee, Ph.D.

Academic Editor

PLOS ONE

Additional Editor Comments (optional):

Reviewers' comments:

Reviewer's Responses to Questions

**Comments to the Author**

1. If the authors have adequately addressed your comments raised in a previous round of review and you feel that this manuscript is now acceptable for publication, you may indicate that here to bypass the “Comments to the Author” section, enter your conflict of interest statement in the “Confidential to Editor” section, and submit your "Accept" recommendation.

Reviewer #1: All comments have been addressed

Reviewer #2: All comments have been addressed

2. Is the manuscript technically sound, and do the data support the conclusions?

Reviewer #1: Yes

Reviewer #2: Yes

3. Has the statistical analysis been performed appropriately and rigorously? 

Reviewer #1: Yes

Reviewer #2: Yes

4. Have the authors made all data underlying the findings in their manuscript fully available?

Reviewer #1: Yes

Reviewer #2: Yes

5. Is the manuscript presented in an intelligible fashion and written in standard English?

Reviewer #1: Yes

Reviewer #2: Yes

6. Review Comments to the Author

Reviewer #1: The author has thoroughly addressed all the comments given and thus I suggest the article be accepted for publication.

Reviewer #2: Thank you for carefully addressing the suggested changes. I believe your review will be of great value for public policies and health interventions.

7. PLOS authors have the option to publish the peer review history of their article (what does this mean?). If published, this will include your full peer review and any attached files.

Reviewer #1: **Yes: **Mario Sibamenya Venance

Reviewer #2: **Yes: **Heber Gómez-Malavé

---

## [Editor Report · Acceptance letter]

4 Feb 2021

PONE-D-20-26914R1 

Availability of adequately iodized salt at the household level in Ethiopia: a systematic review and meta-analysis 

Dear Dr. Alamneh:

I'm pleased to inform you that your manuscript has been deemed suitable for publication in PLOS ONE. Congratulations! Your manuscript is now with our production department. 

Kind regards, 

on behalf of

Dr. Shaun Wen Huey Lee 

Academic Editor

PLOS ONE